# A Study of the Relationship between University Students’ Food Neophobia and Their Tendencies towards Orthorexia Nervosa

**DOI:** 10.3390/bs13120958

**Published:** 2023-11-21

**Authors:** Ayse Gumusler Basaran, Yagmur Demirel Ozbek

**Affiliations:** Faculty of Health Sciences, Recep Tayyip Erdoğan University, Rize 53350, Turkey; ayse.basaran@erdogan.edu.tr

**Keywords:** university student, orthorexia, food neophobia, eating behavior

## Abstract

Food neophobia, known as an avoidance of the consumption of unknown foods, can negatively impact nutritional quality. In orthorexia nervosa, there is an excessive mental effort to consume healthy food. Individuals exhibiting symptoms of food neophobia and orthorexia nervosa may experience food restrictions. This study aimed to assess food neophobia levels and orthorexia nervosa tendencies among university students, investigate the potential association between the two constructs, and explore the effect of the demographic characteristics of the participants on the variables. This is a descriptive cross-sectional study. The study sample consisted of 609 students enrolled at Recep Tayyip Erdoğan University. The data were collected through Google Forms using a sociodemographic information form, the Food Neophobia Scale, and the ORTO-11 scale. Ethics committee approval and institutional permission were obtained for the study. Of the students participating in the survey, 71.9% were female, 14.6% were classified as neophobic, and 47.1% had orthorexia nervosa symptoms. The mean scores from the Food Neophobia Scale (39.41 ± 9.23) and the ORTO-11 scale (27.43 ± 5.35) were in the normal range. Food neophobia was significantly higher among those who did not consume alcohol. Orthorexia nervosa symptoms were significantly more common among married people. In the correlation analysis, no significant relationship was found between age, food neophobia, and orthorexia nervosa. It can be said that food neophobia in this study is similar to in other studies conducted on university students. In addition, about half of the participants had symptoms of orthorexia nervosa. This result is higher compared to other studies conducted with university students. The findings of this study indicate that the participants care about the healthfulness of food.

## 1. Introduction

Nutrition is the intake and utilization of the nutrients and bioactive components that are necessary for physical and mental growth and development, the maintenance of life, the improvement and protection of health, quality of life, and productivity [1]. Being healthy depends on adequate and balanced nutrition [2]. Diversity in food consumption plays a vital role in ensuring sufficient and balanced nutrition [3]. Food choices, in other words, which nutrients will be taken into the body, affect health status [4]. Food choices are affected by many internal and external factors, including food content, food accessibility, family and close circles of friends, social and cultural characteristics, nutritional information, emotional states, and food experiences [4,5]. Studies have found that cultural differences in food choices cause significant changes [6,7]. Incorrect nutrition practices increase the risk of eating behavior disorder [8].

Food neophobia means avoiding tasting or experiencing unusual or new foods/drinks [9]. People may reject fresh foods because they are potentially dangerous [10]. This varies from person to person. On the other hand, increasing interactions between cultures have helped to reduce food neophobia [11]. Factors affecting food neophobia include the food choices of parents and peers, hereditary factors, environment, sex, age, educational status, and place of residence [12,13,14,15]. Food neophobia is more prevalent among preschoolers and older people, while young individuals are more open to trying new foods [16,17]. Research suggests that there is a higher prevalence of food neophobia among males [18,19,20]. Also, educational background affects the propensity toward food neophobia [21]. In addition, it was discovered that the prevalence was more significant among individuals residing in rural regions [22,23]. The research on university students revealed that the food neophobia scores fell within the range of 22–37.3. These scores were considered neutral [13,22,24,25]. Food neophobia differs from country to country and directly reduces diet quality and diversity. Decreased dietary diversity, in turn, leads to the restriction of food intake [26].

Another issue that is frequently mentioned with regard to restricting food intake is orthorexia nervosa [27]. Orthorexia nervosa is defined as an excessive mental preoccupation with healthy eating and healthy foods [28]. The term orthorexia is derived from the Greek words “orthos”, meaning “right, appropriate”, and “orexis”, meaning “appetite” [29]. Orthorexia nervosa was first described by Steven Bratman in 1997 [30]. Nevertheless, it has not been included in psychiatric diagnostic classifications yet. Orthorexia nervosa is characterized by healthy food consumption, focusing on quality rather than the quantity of food, not consuming when the provenance of food is ambiguous, and paying particular attention to food preparation, cooking methods, and food production, processing, and packaging [31,32,33]. Therefore, individuals with orthorexia nervosa may experience feelings of guilt and inadequacy regarding their failure to adhere strictly to healthy nutrition [34], tend to stick to rigorous self-imposed guidelines for meals [32], and consume raw vegetables and fruit [33]. Moreover, individuals with orthorexia nervosa may experience trace element deficiencies, anxiety about criticisms of their eating habits, and social isolation due to dietary restrictions [33]. These factors negatively affect physical health, interpersonal relationships, stress management, and mental health [34]. Studies carried out in different countries have reported orthorexia prevalence in the range of 76.7% to 1.7% among university students [10,11,23,35,36,37,38,39,40]. On the other hand, several studies have reported a lack of statistically significant difference between sexes in terms of the prevalence of orthorexia nervosa [3,11,23,36,37]. In some studies, the incidence of orthorexia nervosa was found to be higher in young individuals. Studies conducted with students determined that married people and smokers had more orthorexia symptoms [23,30]. The possibility of affecting the eating behavior of other family members for married people makes the issue important.

University students’ healthy eating behaviors are affected by various factors, such as time constraints, unhealthy snacking, fast food, stress, the high prices of healthy food, and easy access to junk food [41]. In addition, changes in residence, unfavorable financial situations, and living alone or with friends in student dormitories or apartments or with other families lead to significant changes in the lifestyles of young people. For this reason, it is observed that eating disorders are increasing, especially among university students [42]. According to one study, 78.9% of university students think that they need a healthy diet. The same study also found that students often skip meals, and that the frequency of milk, fruit, and vegetable consumption among students is lower than expected. [43]. For this reason, university students’ eating attitudes are becoming an important issue that needs to be investigated.

The incidence of food neophobia and orthorexia nervosa, which are two terms with an important place in food restriction, varies according to demographic characteristics. Nevertheless, the association between demographic factors and the occurrence of food neophobia and orthorexia nervosa remains inconclusive. Determining the relationship between these two eating behaviors is essential regarding nutritional limitations and vitamin and mineral deficiencies. In addition, irregular nutrition in university students makes nutritional restriction more critical. In this regard, this study will make a valuable contribution to the existing body of literature. The primary purpose of this study was to investigate the levels of food neophobia and orthorexia nervosa tendencies among university students, as well as to assess how the two constructs correlate to demographic characteristics. The secondary purpose of this study was to determine the correlation between food neophobia and orthorexia nervosa.

## 2. Materials and Methods

### 2.1. Sample of the Study

This research, conducted in 2021, is a descriptive, cross-sectional study. The study population comprised 15,742 undergraduate and associate degree students currently enrolled at Recep Tayyip Erdoğan University. The study sample consisted of 609 university students who volunteered to participate. The sample size was calculated utilizing the Raosoft Sample Size Calculator program. To determine the sample size, an analysis was conducted on recent studies carried out in Turkey on orthorexia nervosa and food neophobia among students. In their research, Garipoğlu et al. observed the prevalence rate of orthorexia nervosa as 76.7% among the student population [35]. When this ratio, 99% reliability, and a 5% maximum error were considered, the sample size was determined as n = 457. At the same time, in the study conducted by Palamutoğlu et al., the prevalence of food neophobia was found to be 22.6% [44]. Considering this prevalence, the value of n was established as 451. The minimum sample size was calculated, and the study was completed with 609 students

### 2.2. Collection of Data

After ethics committee approval and institutional permission, the research questionnaire was sent to the students’ institutional e-mail addresses. A link was created via Google Forms that included the survey items, information about the study, and the students’ approval. It took approximately 10 min to fill out the survey. No payment was made to the participants. Six hundred nine students voluntarily agreed to participate in the study and completed the survey.

### 2.3. Data Collection Tools

The research data were collected using the sociodemographic information form developed by the researcher, the Food Neophobia Scale, and the ORTO-11 scale.

#### 2.3.1. Sociodemographic Information Form

The form consisted of 11 items about the participants’ age, sex, marital status (single, married), education years (1st-year students, 2nd-year students, 3rd-year students, and 4th-year students), the type of high school graduated from (health vocational high school, other) smoking status (smoker, non-smoker, and former smoker), alcohol consumption status (yes and no), number of meals, place of residence (rural and urban), presence of food allergy, and person/place providing new food recommendations (family, friends, social media, and does not take suggestions). The participants’ age, gender, number of meals, and food allergies were asked as open-ended questions on the form.

#### 2.3.2. Food Neophobia Scale

The Food Neophobia Scale (FNS) was used to assess the tendency of individuals to avoid or try new foods. The Turkish validity and reliability of the scale developed by Pliner and Hobden were tested by Duman [45,46]. The 10-item 7-point Likert-type scale is scored in the range of 10–70, with high scores indicating food neophobia and low scores indicating enjoyment in trying new foods. The food neophobia classification was made based on the mean ± standard deviation ($\bar{x}$ ± SD). Individuals with an FNS score of <$\bar{x}$ − SD were considered to enjoy fresh foods/neophilic, $\bar{x}$ ± SD to be neutral, and >$\bar{x}$ + SD to be highly fearful of new foods/neophobic [13,47]. The Cronbach’s alpha internal consistency coefficient for the integrity of Duman’s FNS was 0.614. In the present study, the Cronbach’s alpha value was 0.642.

#### 2.3.3. Orthorexia Nervosa-11 (ORTO-11) Scale

The Orthorexia Nervosa Scale, developed by Steven Bratman, is a 10-item scale designed to assess healthy-eating obsession symptoms in individuals. Donini et al. developed a 15-item ORTO-15 scale by adding some questions [48]. Its validity and reliability in Turkish was assessed by Arusoğlu in 2006, and the Cronbach’s alpha value was calculated as 0.44. Due to the low Cronbach’s alpha value, Arusoğlu re-evaluated the same scale in 2008 and removed questions 1, 2, 9, and 15 from the scale [30]. Each statement in the ORTO-11 scale is evaluated with a 4-point Likert-type rating (1: always, 4: never), and only the 8th item is reverse-coded. The Cronbach’s alpha value of the ORTO-11 scale was found to be 0.701. Fidan et al. determined 27 points on the ORTO-11 scale as the threshold for sensitivity to orthorexia nervosa; in other words, <27 points indicated the presence of orthorexia nervosa [20]. In our study, a score of <27 was accepted as an indicator of orthorexia nervosa.

### 2.4. Analyzing the Data

The statistical data analysis was conducted using the SPSS 22 software package. Descriptive data were expressed as percentages, means, and standard deviations. The Kolmogorov–Smirnov test was used for the normality distribution of the data. The Mann–Whitney U test was used to compare the two groups. Kruskal–Wallis, Bonferroni correction, and Tamhane’s T2 post hoc tests were used to compare three or more groups. Spearman’s correlation analysis was used to examine the relationship between the scales. In the correlation analysis, 0–0.39 was considered a weak correlation, 0.40–0.69 a moderate correlation, 0.70–0.89 a strong correlation, and 0.90–1.00 a robust correlation. The significance value was accepted as *p* < 0.05.

### 2.5. Ethical Dimension

This research was conducted with the approval of the Social and Humanities Ethics Committee of Recep Tayyip Erdoğan University (2021/152). Permission was also obtained from the university.

## 3. Results

Of the participants, 28.1% were male and 71.9% female. Additionally, 94.6% reported being single. The participants’ average age was 22.33 (SD = 4.96) years; 29.4% of the students were first-year students, 34.3% second-year, 16.7% third-year, and 19.5% fourth-year. A total of 10.5% of the students were graduates of a health vocational high school. Additionally, 23.6% of the students resided in rural areas, and 16.1% were smokers, whereas 77.5% of the students reported having never smoked and 6.4% reported having quit smoking. Furthermore, 7.1% of the students reported consuming alcohol, and 11.8% reported having a food allergy. According to the survey data, 54.7% of the students stated that they had two main meals, and 5.1% had one. Moreover, 37.8% had one intermediate meal, and 32.8% had two. While 30.5% of students received new food suggestions from their friends, 33.3% stated they did not receive any new food recommendations. The prevalence of food neophobia was found to be 14.6%, while 15.9% were found to be neophilic. On the other hand, the prevalence of orthorexia nervosa was 47.1%. Descriptive data are shown in Table 1.

Considering the sexes, while 47.9% of females had a risk of orthorexia, 45% of males had a risk of orthorexia. While 17.4% of females had neophobia, 7.6% of males had neophobia. The distribution by sex is shown in Figure 1.

The participants’ scores from the FNS and the ORTO-11 scale are shown in Table 2, and the relationships of the scores with independent variables are shown in Table 3

The mean FNS score was 39.41 (SD = 9.23), while the mean ORTO-11 scale score was 27.43 (SD = 5.35). The participants’ mean FNS score was neutral, while their mean ORTO-11 scale score was in the normal range.

The FNS score was significantly higher in those who did not consume alcohol (*p* = 0.044). In addition, the person/place providing new food recommendations greatly affected food neophobia (*p* = 0.024). However, there was no significant difference between the groups after the Bonferroni correction. Sex (*p* = 0.678), marital status (*p* = 0.629), place of residence (*p* = 0.081), smoking status (*p* = 0.244), food allergy status (*p* = 0.283), or the number of primary (*p* = 0.461) and intermediate meals (*p* = 0.214) did not create a significant difference in food neophobia (Table 3).

There was a statistically significant association between marital status and orthorexia nervosa symptoms: the symptoms were more prevalent among married students (*p* = 0.007). Furthermore, there was a statistically significant association between the consumption of intermediate meals (*p* = 0.034) and the provision of new food recommendations (*p* = 0.048), with individuals exhibiting elevated levels of orthorexia nervosa symptoms. In the post hoc analysis, those who consumed no intermediate meals and those who consumed two intermediate meals obtained significantly higher risk of orthorexia than those who consumed four intermediate meals. However, there was no significant difference between the groups after the Bonferroni correction. Additionally, those who did not receive food recommendations from anyone obtained significantly higher scores than those who received food recommendations from social media. However, there was no significant difference between the groups after the Bonferroni correction. Sex (*p* = 0.760), place of residence (*p* = 0.054), alcohol consumption (*p* = 0.378), smoking status (*p* = 0.753), food allergy status (*p* = 0.205), or the number of main meals (*p* = 0.695) did not create a significant difference in orthorexia (Table 4).

The correlation analysis found no significant relationship between age, food neophobia (r = 0.002, *p* = 0.970), and orthorexia (r = −0.073, *p* = 0.070). Moreover, no meaningful relationship was found between food neophobia and orthorexia (r = −0.016, *p* = 0.698) (Table 4).

## 4. Discussion

Food choice has been the subject of many studies in recent years [9,12]. This study was conducted to evaluate the food neophobia and orthorexia tendencies of university students. The prevalence of food neophobia among students was determined to be 14.6%. In Turkey’s population of university students pursuing a degree in health sciences, the prevalence rates were 22.6% and 13.5% [22,44]. This difference may be due to the difference in departments, since the study included participants from fields other than health sciences.

The study’s mean FNS score was 39.41 (SD = 9.23) (neutral). This score was reported as 37.3 in the Turkish population [22], 29.8 in the American population [13], 36.4 in the Lebanese population [13], 22.0 in the Italian population [24], and 33.6 in the Chinese population [25]. There may be variations in the mean FNS score across different populations. Although previous research revealed higher rates among the American population and Italian youth, it can be argued that these rates are comparable to those observed in other countries. Different food preparation and consumption patterns in different countries may affect food neophobia.

Consistent with previous research, no statistically significant associations were found between gender [21,22,49,50], age [12,51], and food neophobia. However, in terms of gender, some studies found that food neophobia was more prevalent in men [18,19,20], while one study observed a higher prevalence among women [52]. The findings of the studies show a degree of inconsistency about gender. In contrast, in terms of age, several studies have shown that food neophobia increases with age [53,54,55]. This difference may be due to the different age ranges of the people in the study. In this study, the ages of the students were close to each other, and the age range was narrow.

This study revealed that residing in rural or urban areas did not yield a statistically significant disparity concerning food neophobia. However, individuals living in rural areas exhibited a higher food neophobia score. Prior research has underscored a notable gap in food neophobia between rural and urban areas, with a higher prevalence observed in the former and a decrease observed as urbanization progresses [22,23]. The lower prevalence of food neophobia among urban residents may be due to more interaction with different cultures and easier access to new foods.

The observed prevalence of food neophobia among individuals who consumed alcohol in the present study aligns with the findings reported by Aiello et al. [24]. The lack of a significant association between educational background and food neophobia is consistent with the results obtained from a study conducted among university students in Mersin [56]. The fact that individuals who consume alcohol are more open to cultural interaction may have reduced their food neophobia [16]. Individuals who do not drink alcohol because they think it is harmful to their health may also have concerns about the effects of new foods on their bodies, which may have caused the high rate of food neophobia among them. Steps such as introducing the ingredients of fresh foods and planning tasting menus can be taken to reduce food neophobia.

In this particular study, the proportion of participants with orthorexia nervosa symptoms was 47.1%. The mean ORTO-11 scale score was 27.4 ± 5.35. However, a higher prevalence, between 59.2 and 76.7%, has been reported in other studies conducted with nutrition and dietetics students [11,35,37]. While the mean ORTO-11 scale was found to be 27.3 ± 4.53, 26.3 ± 4.9, and 26.8 ± 6.24 in other studies conducted with students [20,57,58], it was found to be higher (38.23 ± 3.28) in dietetics students [39]. The fact that students in different departments have different levels of knowledge about nutrition and health may have reduced the possibility of people with higher levels of orthorexia nervosa symptoms. The prevalence of orthorexia nervosa symptoms among university students varies across different countries, with 74.5% in Lebanese [10], 31.2% in Italian [40], 25.2% in Spanish [38], and 1.7% in Bangladeshi people [23]. These rates illustrate variations in cultural norms and practices. This study revealed that approximately 50% of the students exhibited tendencies indicative of orthorexia nervosa. The reason for this may be the importance given to health due to the COVID-19 pandemic that started at the end of 2019. It is thought that the tendency for healthy eating may have increased after the pandemic. In addition, students who do not have the opportunity to prepare their meals have to eat in canteens or collective cafeterias, which may have affected this. Furthermore, with the influence of popular culture on students and the impact of students on each other, consuming healthy food can become an obsession over time. This may cause an increase in prevalence over time. Increasing symptoms of orthorexia nervosa can gradually threaten an individual’s quality of life. To prevent this, students should be provided with information about eating disorders and healthy food choices. Additionally, universities may offer courses that include nutrition information to students as elective courses.

The present study’s finding that married students had more orthorexia nervosa symptoms is similar to other studies [23,30]. This may be because married people tend to care about their family members’ health. Married participants may influence the eating behavior of other family members. Information studies are needed to prevent married participants showing orthorexia symptoms from negatively affecting their family members. In individuals exhibiting symptoms of orthorexia, the causes should be investigated, and solutions should be devised accordingly.

In this study, the finding that gender [3,11,23,36,37], age [11,59,60], and place of residence [23] did not make a significant difference in people with higher orthorexia nervosa symptoms is similar to other studies. However, in some studies, the ORTO-11 scale score was higher in women [11,30] than men [20,48]. It can be concluded that these studies reported inconsistent findings in terms of gender. Similarly, some studies suggest a weak negative [61] or weak positive [62] relationship between age and orthorexia nervosa. This difference may be due to the different age distribution of participants or environmental factors.

The finding that smoking status did not make a difference in higher levels of orthorexia nervosa symptoms in this study is similar to that of Fidan’s study [20]. However, there are studies in which orthorexia was found to be higher in smoking [23] and non-smoking students [36]. This shows that there is uncertainty about the effect of smoking on the prevalence of orthorexia nervosa. Studies may have produced different results due to the duration of cigarette consumption and its impact on health.

This study found no significant relationship between mean ORTO-11 scores and FNS scores. The same finding was also reported by other studies [3,47]. This suggests that individuals with a healthy-eating obsession may not have food neophobia, and it does not affect their desire to consume. If the food is thought to be healthy by students with orthorexia symptoms, it is possible to consume it even if it is unknown. On the other hand, students with food neophobia may not consume foods they do not know about, even if they are healthy.

Limitations: As the sample includes students, it has a small age range. The number of male participants is lower than that of female participants. In addition, students are less likely to reside in rural areas than in urban areas. Also, the number of married students is lower than that of single students. However, this study also has its strengths: it includes students from different departments, the number of participants is not typical, and it addresses two topics together. These research findings can be generalized to university students. They reflect the youth in Turkey.

## 5. Conclusions

In this study, investigating university students’ food neophobia and orthorexia nervosa tendencies, 14.6% of the students were neophobic and 47.1% had higher orthorexia nervosa symptoms. In addition, food neophobia was higher in those who did not consume alcohol. Moreover, orthorexia nervosa symptoms were found to be more prevalent in married people. Also, no significant relationship was found between food neophobia and orthorexia.

Consequently, the prevalence of food neophobia aligns with anticipated levels when accounting for other relevant research findings. The prevalence of orthorexia nervosa was observed to be greater than that reported in numerous previous studies. This phenomenon could be attributed to the pandemic during the study period, which resulted in a heightened emphasis on matters of health. Conducting similar studies in the future will shed light on the impact of the pandemic. Given the potential consequences of this elevated rate, such as food restriction and the potential for the development of additional eating disorders, action must be taken. Educating university students on healthy nutrition and eating disorder awareness is imperative. This is because food restriction can pave the way for severe diseases in the long term. Furthermore, it is feasible to acquire knowledge regarding nutritional composition by utilizing novel applications facilitated by technological advancements. However, all stakeholders in the food sector must be transparent and share information in a way that allays people’s fears. The results of this study will provide a framework for future investigations on the prevalence of research topics and the relationship between them. The literature contains conflicting results about orthorexia nervosa with regard to both gender and age. Therefore, it may be recommended to conduct multicenter studies in large populations involving different age groups. In addition, it would be helpful to run qualitative research methods on the reasons for a fear of new foods and the reasons for developing healthy-eating obsessions.

## Figures and Tables

**Figure 1 behavsci-13-00958-f001:**
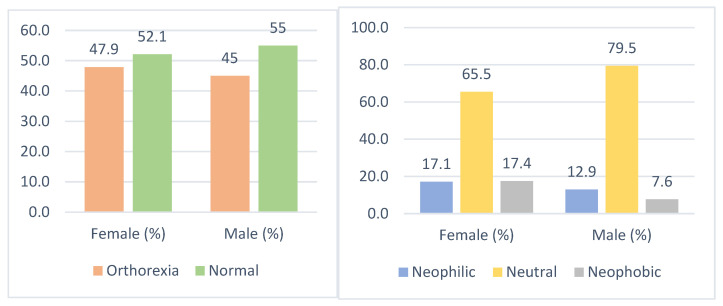
Distribution of food neophobia and orthorexia nervosa by sex.

**Table 1 behavsci-13-00958-t001:** Descriptive data of the participants.

Independent Variables		N	%
Age $\bar{x}$ (SD)	22.33 (SD = 4.94)		
Sex	F	438	71.9
	M	171	28.1
Marital status	Single	576	94.6
	Married	33	5.4
Place of residence	Rural	144	23.6
	Urban	465	76.4
Smoking status	Smoker	98	16.1
	Non-smoker	472	77.5
	Quit	39	6.4
Alcohol consumption	Drinks	43	7.1
	Does not drink	566	92.9
Number of Main meals	1	32	5.3
	2	333	54.7
	3	227	37.3
	4	17	2.8
Number of Intermediate meals	0	92	15.1
	1	230	37.8
	2	200	32.8
	3	66	10.8
	4	21	3.4
Person/place providing new food recommendation	Family	133	21.8
	Friend	186	30.5
	Social media	87	14.3
	Does not take suggestions	203	33.3
Food Neophobia	Neophilic (≤30)	97	15.9
	Neutral (31–48)	423	69.5
	Neophobic (≥49)	89	14.6
ORTO-11	Normal (>27)	322	52.9
	Orthorexia risk (≤27)	287	47.1

**Table 2 behavsci-13-00958-t002:** Participants’ mean scores from the FNS and the ORTO-11 scale.

Scales	N	Min–Max	$\bar{x}$	SD
Food Neophobia	609	10–67	39.41	9.23
ORTO-11	609	14–41	27.43	5.35

**Table 3 behavsci-13-00958-t003:** Comparison of scores from the FNS and the ORTO-11 scale according to some characteristics of participants.

Independent Variables		N	Food Neophobia	*p*	ORTO-11	*p*
			Mean Rank		Mean Rank	
Sex	Female	438	306.85	0.678	303.64	0.760
	Male	171	300.27		308.47	
			U = 36640.5, Z = −0.415		U = 36855.0, Z = −0.305	
Marital Status	Single	576	305.82	0.629	309.61	0.007
	Married	33	290.61		224.6	
			U = 9029.0, Z = −0.484		U = 6851.0, Z= −2.704	
Place of residence	Rural	144	327.35	0.081	329.67	0.054
	Urban	465	298.08		297.36	
			U = 30261.0, Z= −0.747		U = 29928.0, Z= −1.929	
Alcohol Consumption	Drinks	43	252.87	0.044	282.23	0.378
	Does not drink	566	308.96		306.73	
			U = 9927.5, Z= −2.017		U = 11190.0, Z= −0.882	
Smoking status	Smoker	98	283.08	0.244	310.01	0.753
	Non-smoker	472	311.44		305.57	
	Quit	39	282.10		285.4	
			KW X^2^ = 2.821		KW X^2^ = 0.566	
Food allergy	Yes	72	325.87	0.283	280.35	0.205
	No	537	302.20		308.30	
			U = 17829.5, Z = −1.073		U = 11557.5, Z = −1.268	
Number of Main meals	1	32	312.42	0.461	334.30	0.695
	2	333	311.74		307.78	
	3	227	298.21		297.64	
	4	17	249.53		293.56	
			KW X^2^ = 2.579		KW X^2^ = 1.444	
Number of Intermediate meals	0	92	319.03	0.214	281.33	0.034
	1	230	313.47		306.77	
	2	200	305.43		296.58	
	3	66	258.77		323.94	
	4	21	292.00		410.07	
			KW X^2^ = 5.801		KW X^2^ = 10.437	
Person/place providing new food recommendation	Family	133	219.52	0.024	308.03	0.048
	Friend	186	312.90		310.97	
	Social media	87	251.96		256.94	
	Not taking suggestions	203	311.03		318.15	
			KWX^2^ = 9.475		KWX^2^ = 7.907	

U, Mann–Whitney U; Z, Z-statistics; KW, Kruskal–Wallis; *p*, *p*-value; n, number of people.

**Table 4 behavsci-13-00958-t004:** Spearman correlation coefficients between age, FNS, and ORTO-11 scale.

		Age	Food Neophobia	ORTO-11
Age	r	1	0.002	−0.073
	*p*		0.970	0.070
Food neophobia	r		1	−0.016
	*p*			0.698
ORTO-11	r			1
	*p*			

## Data Availability

Data will be available upon genuine request from the corresponding author.

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
