# Peer review of "A Study of the Relationship between University Students’ Food Neophobia and Their Tendencies towards Orthorexia Nervosa"

_behavsci, 2023, doi:10.3390/bs13120958_

Round 1
Reviewer 1 Report (New Reviewer)
Comments and Suggestions for Authors
In the present study, the authors investigate the levels of food neophobia and orthorexia nervosa tendencies among university students and assess how the two constructs correlated to demographic characteristics. In addition, the correlation between food neophobia and orthorexia nervosa was assessed. In my opinion, the research topic is meaningful, but there are still some errors in this paper which need to be modified.
Abstract
In line 14, “and explore the effect of demographic characteristics on the variables.” The research object of this paper is university students, whether the sample of university students can represent the “demographic characteristics”.
In line 27-28, ”This situation reveals that, in addition to food-related training, professionals should inform the public about food safety.” The research object of this paper is university students, this sentence can be changed to the opinions put forward to participants in view of the survey results may be better.
Introduction
In line 59-60,“The research conducted on university students revealed that the food neophobia score fell within the range of 22-37.3” Please describe in detail what is the result represented by this score.
There are some errors in the grammatical tense of the second paragraph of the introduction, please modify it.
In line 78-79, “Studies carried out in different countries have reported orthorexia prevalence in the range of 76.7% to 1.7% among university students” , the research results of these references are too different.
Materials and Methods
In line 115-116,“The sample of the 115 study consisted of 609 students who agreed to participate in the study”. According to the article, the choice of study samples is determined by personal willingness, whether there are other screening criteria.
Result
The first line of the first paragraph of the result section should be indented by two characters. Please check the full text to see if there are similar problems.
Please keep the font in Figure 1. consistent with the fonts of other parts of the article.
The title of Table 4. is not in the correct position. Please amend
In line 232, the writing format of” (r= 0.002, p= 0.970)”and”(r= .073, p= .070)” is not uniform. Please check whether there are similar errors in the full text and unify the writing format.
In line 257-260,the sample selected in this study is university students, and the sample age range is small. I think the results of this study are not enough to prove that there is no significant correlation between age and food neophobia.
Comments on the Quality of English Language
Moderate editing of English language required.
Author Response
Please see the attachment.

Reviewer 2 Report (New Reviewer)
Comments and Suggestions for Authors
The study aimed at assessing the prevalence of neophobia and orthorexia among students from a Turkish university, as well as the potential association between the two concepts/attitudes, together with the influence of some demographic characteristics on the two parameters.
The methodology used is described with enough details. The sample size is justified and sufficient. The results are presented in a logical manner. The results are discussed and compared with findings from other studies. Some identified limitations are mentioned in the manuscript. The stated conclusions are supported by results.
The manuscript is generally well written and in a coherent manner.
Author Response
Please see the attachment.

Reviewer 3 Report (New Reviewer)
Comments and Suggestions for Authors
This article represents the study on the relationship between food neophobia and orthorexia nervosa tendencies among university students, as well as the potential impact of demographic characteristics on these variables. While the research addresses an important issue concerning eating behaviors and nutrition, there are several critical points to consider:
1. The abstract lacks context and background information on both food neophobia and orthorexia nervosa. While these terms are mentioned, it would be beneficial to provide a brief explanation or definition to ensure the reader understands the study's focus. The abstract could benefit from improved clarity and organization of information. Important details, such as the methodology and main findings, should be presented in a more structured and coherent manner.
2. The results demonstrated that approximately half of the participants had orthorexia nervosa symptoms, yet the study does not delve into the potential consequences of this finding. The study's implications for public health and nutrition could be discussed in more detail.
3. The results and main conclusions of the study on university students' food neophobia and orthorexia nervosa tendencies provide valuable insights into the prevalence of these eating behaviors and their potential implications. The observation that food neophobia is higher in individuals who do not consume alcohol and that orthorexia nervosa symptoms are more prevalent in married people is interesting but lacks a clear explanation in the manuscript. It would be useful to discuss potential reasons for these associations and how they might inform intervention strategies or public health efforts.
4. The manuscript rightly emphasizes the need for action in response to the elevated prevalence of orthorexia nervosa, including education on healthy nutrition and eating disorder awareness. The manuscript does not discuss potential areas for further research, which could be beneficial in guiding the research community toward addressing gaps in our understanding of food neophobia and orthorexia nervosa. It would be helpful to specify what kind of research questions or interventions could be useful.
5. In conclusion, the manuscript provides information on the prevalence of food neophobia and orthorexia nervosa tendencies among university students. However, it would benefit from more detailed analysis, contextual information, and specific recommendations to fully understand and address the implications of these findings. Although the link between the pandemic and orthorexia nervosa is a plausible hypothesis, it is speculative and should be treated with caution and supported by further research.
I also have some minor suggestions for the authors.
1. The authors should reformulate the Introduction part (lines 32-45; 48-52). The sentences are complicated to read.
2. Reformulate and clarify the sentence in lines 59-60.
3. Line 75…did you mean trace element deficiencies or something else? Please, specify.
4. Line 86… convenience high-calorie food…please, clarify.
5. Line 88….city change…did you mean change of residence? Reformulate, please.
6. Line 92-93, please, reformulate the sentence.
7. Correction in Figure 1 is needed. Neophobic.
Author Response
Please see the attachment.

Reviewer 4 Report (New Reviewer)
Comments and Suggestions for Authors
Please see the attachment.

Comments on the Quality of English Language
English proofreading is suggested.
Author Response
Please see the attachment

Reviewer 5 Report (New Reviewer)
Comments and Suggestions for Authors
Thank you for inviting me to review this manuscript. I consider that it is well structured and thorough. I was not strongly persuaded by the rationale for the study. I feel that there are so many influences on students' diets that I question the value of measuring these two conditions. However, I accept the general principle of generating new knowledge.
I found the discussion section rather vague. I would recommend greater clarity and/or a more compelling argument about what this study tells us and how we will benefit from this new knowledge.
Round 2
Reviewer 3 Report (New Reviewer)
Comments and Suggestions for Authors
The authors have tried to address all the raised issues. However, after re-reading the entire text, my impression is that it lacks scientific soundness. I've noticed that the English language usage and sentence construction need some attention to enhance clarity and readability. The sentences are unclear, incoherent, complicated; the use of tenses should be checked. Unfortunately, the manuscript sounds like a newspaper article in certain segments. I believe addressing these language issues will greatly improve the overall quality of the article. This can be seen in lines 270-275; 289-298; 308-313 and 366-370. The sentence in lines 366-367 is too general, please specify.
Comments on the Quality of English Language
. I've noticed that the English language usage and sentence construction need some attention to enhance clarity and readability. The sentences are unclear, incoherent, complicated; the use of tenses should be checked. Unfortunately, the manuscript sounds like a newspaper article in certain segments. I believe addressing these language issues will greatly improve the overall quality of the article.
Author Response
Please see the attachment

Reviewer 4 Report (New Reviewer)
Comments and Suggestions for Authors
I would like to thank the authors for their efforts to improve the manuscript.
Some rearrangements have been made in the text, but there are still some things that should be clarified or changed.
Can you make such a strong statement: “Orthorexia nervosa symptoms were significantly more common in married individuals“ when less than 6% of the population was married?
The last sentence of the abstract is not part of the main aim and scope of the manuscript, and if read independently, the link between orthorexia, neophobia and additives is not clear
Again, there is reference “Ministry” which has already been classified as questionable, especially as the sentence is very similar to the sentence from the source Meral G (2018) Philosophy of Nutrition: Past-Future Nutrition. Gut Microbiota - Brain Axis. IntechOpen. Available at: http://dx.doi.org/10.5772/intechopen.80726.
„Nutrition is an action that must be done consciously to get the nutritional items needed by the body in sufficient quantities and at the right time to maintain health and improve quality of life.“
Nutrition is an action that must be carried out consciously in order to obtain the nutrients required by the body in sufficient quantities and at the right time to maintain health and improve quality of life.
Please have the manuscript proofread by a native speaker
Comments on the Quality of English Language
The manuscript contains linguistic and grammatical inconsistencies, so I recommend having the manuscript proofread by a native speaker.
Author Response
Please see the attachment

This manuscript is a resubmission of an earlier submission. The following is a list of the peer review reports and author responses from that submission.
Round 1
Reviewer 1 Report
Comments and Suggestions for Authors
The manuscript presents an interesting and relevant research. The introduction provides adequate background and overall the methods allow replication and the results are adequatelly presented and discussed. The following (mostly minor) issues should be addressed:
1. Please avoid terms like "impact" (line 11), as the design does not allow to draw causal relationships.
2. Delete "by email" in line 13.
3. When reporting standard deviations, please correct to "SD = xxx", and the "+-" sign may indicate other dispersion measures. E.g. in the abstract: 39.41 (SD = 9.23).
4. Sentence in lines 17-18 refers to the mean values but is formulated as all participants had scores in the normal range; as the % of participants with high neophobia and ON symptoms is previously reported, I suggest deleting this sentence.
5. Please replace "gender" (a complex and non-dichotomous social construct) with "sex" (male/female) througout the manuscript.
6. In line 42 (FN score in the range of 22 to 37.3) please refer the instrument used.
7. In line 62 replace "and the prevalence" with "in the prevalence" (the comparison is not between gender/sex and the prevalence, but between males and females, regarding prevalences).
8. Rephrase the sentence in lines 63-65, as the studies in the first and second part of the sentence are not the same.
9. Line 66: replace "correlation" with "relationship".
10. Line 82: "population" instead of "sample".
11. Line 91: replace "When the value" with "Considering this prevalence, the value".
12. In lines 109-110, please correct the ranges: "Individuals with a FNS score < (X - SD) are [...] and > (X + SD) as [...]".
13. Line 118: "and removed questions 1, 2, 9 and 15 from".
14. Line 138: replace "smoking" with "smoked" or "were smokers".
15. Rephrase "Living in areas" in text and tables.
16. Line 146: replace "comparisons" with "relationships".
17. Please rewrite the paragraph in lines 159-166, indicating the direction of the relationships.
18. From the same paragraph, the results regarding some variables are nor presented in the table (living in areas, food alergy, main meals); please add them.
19. In table 3 I suggest to present the medians instead of the mean ranks.
20. Since in table 4 only 6 values matter, these can be referred in the text (lines 171-172), deleting this table.
21. Please rephrase the first sentence of the discussion, as not exactly nutrition includes hunger, satiety, selective eating... Perhaps "eating behaviour"?
22. It is unclear what the two values (22.6% and 13.5%) in lines 186-187 refer to.
23. When referring results in the discussion, please make clear when talking about mean values (e.g. lines 189-191).
24. Line 194: please make clear what kind of cultural interaction.
25. Line 215: this prevalence does not refer exactly to ON symptoms.
26. Line 224: replace with "the reason for this situation may be the ...".
27. Line 227: replace with "married students had higher levels ...".
28. The discussion on the (absence of) relationship between FN and ON (lines 231-234) could be improved in terms of interpretation and implications.
29. Delete the sentence in lines 237-240 ("In this context...") as it is not specific for this study nor should be in the conclusions.
Comments on the Quality of English Language
Included in the overall comments.
Reviewer 2 Report
Comments and Suggestions for Authors
The article dealt with a current topic. Defects are essentially not recognizable. It would be desirable to include illustrations as they make it easier to read.
Reviewer 3 Report
Comments and Suggestions for Authors
Minor English editing may be required at some places, clear run-on sentences and break the long sentences into two or more:
see Lines 209-210. The observed prevalence of food neophobia among individuals who consumed alcohol in the present study aligns with the findings reported by Aiello et al.[16]; Furthermore...
rewrite this sentence: Several factors have been found to have a detrimental impact on physical health, interpersonal relationships, stress management, and mental 58 health[21].
authors should write name of university and body which approved the study must clearly mention..
The research was conducted with the approval of the social and humanities ethics committee of a university (2021/152). Permission was then obtained from the university.
Clearly mention limitations and strength of this study
conclusions should be more clear and relevant looking like a logical end of discussion section
In conclusion authors say that the results of this study will provide a framework for future investigations in the field. But How?
Comments on the Quality of English Language
Minor grammer and editing corrections are required.
